# Recent Shrinkage and Fragmentation of Bluegrass Landscape in Kentucky

**Bo Tao [1], Yanjun Yang [1], Jia Yang [2], Ray Smith [1], James Fox [3], Alex C. Ruane [4], Jinze Liu [5] and Wei Ren [1,*]**

1 Department of Plant and Soil Sciences, College of Agriculture, Food and Environment, University of Kentucky, KY 40546, USA; bo.tao@uky.edu (B.T.); Yanjun.Yang@uky.edu (Y.Y.); raysmith1@uky.edu (R.S.)
2 Department of Forestry, Mississippi State University, MS 39762, USA; jy576@msstate.edu
3 Civil Engineering Department, University of Kentucky, Lexington, KY 40506, USA; james.fox@uky.edu
4 Goddard Institute for Space Studies, National Aeronautics and Space Administration, New York, NY 10025, USA; alexander.c.ruane@nasa.gov
5 Laboratory for High Performance Scientific Computing and Computer Simulation, Department of Computer Science, University of Kentucky, Lexington, KY 40506, USA; jinze.liu@uky.edu
* Correspondence: wei.ren@uky.edu; Tel.: +1-859-257-1953

**Abstract:** The Bluegrass Region is an area in north-central Kentucky with unique natural and cultural significance, which possesses some of the most fertile soils in the world. Over recent decades, land use and land cover changes have threatened the protection of the unique natural, scenic, and historic resources in this region. In this study, we applied a fragmentation model and a set of landscape metrics together with the satellite-derived USDA Cropland Data Layer to examine the shrinkage and fragmentation of grassland in the Bluegrass Region, Kentucky during 2008–2018. Our results showed that recent land use change across the Bluegrass Region is characterized by grassland decline, cropland expansion, forest spread, and suburban sprawl. The grassland area decreased by 14.4%, with an interior (or intact) grassland shrinkage of 5%, during the study period. Land conversion from grassland to other land cover types has been widespread, with major grassland shrinkage occurring in the west and northeast of the Outer Bluegrass Region and relatively minor grassland conversion in the Inner Bluegrass Region. The number of patches increased from 108,338 to 126,874. The effective mesh size, which represents the degree of landscape fragmentation in a system, decreased from 6629.84 to 1816.58 for the entire Bluegrass Region. This study is the first attempt to quantify recent grassland shrinkage and fragmentation in the Bluegrass Region. Therefore, we call for more intensive monitoring and further conservation efforts to preserve the ecosystem services provided by the Bluegrass Region, which has both local and regional implications for climate mitigation, carbon sequestration, diversity conservation, and culture protection.

**Keywords:** grassland shrinkage; fragmentation; urban sprawl; cropland expansion; Kentucky

## 1. Introduction

Globally, land use and land cover changes have reshaped the landscape and substantially altered the capacity of ecosystems to provide various services, such as food and water production, soil fertility, nutrient cycling, biodiversity, socioeconomic and cultural benefits, etc. [1,2]. In the U.S., substantial land use change occurred in the late 19th century and the first half of the 20th century and was characterized by agricultural expansion in the Midwestern U.S. and secondary forest regrowth in the Eastern and Central regions [3]. The total cropland areas in the U.S. peaked in 1940 and showed a

slightly decreasing trend in the following years [4]. Some studies have argued that this decline could also be a result of the changes in the definition of cropland in the USDA census of 1945 and subsequent years [3]. However, there are significant geographic and temporal variations in land-cover change across the U.S. characterized by a mix of expanding, contracting, and stable land use and land cover types across regions [5]. Prior studies have put more emphasis on agriculture and forest-dominated regions such as the Midwest and Southeast U.S., where dramatic land use changes have occurred over the past decades. Recent land use changes in the Central U.S. have received relatively less attention, especially at local scales.

Notwithstanding the importance of changes in land use and cover, fragmentation and loss of natural habitats can disrupt some ecosystems in the same way as loss of water or interior biodiversity. Fragmentation refers to the alteration of previously continuous habitat/landscape into smaller, more isolated patches driven by either natural or anthropogenic disturbances [6,7]. It is tightly linked to land use change and is one of the most pervasive effects of human activities [8]. Generally, three interrelated processes occur when fragmentation happens, i.e., a decrease in the amount of the original vegetation, subdivision of the remaining vegetation into fragments, remnants or patches, and land conversion from the original vegetation to other land use types [9]. These processes decrease landscape connectivity, increase genetic isolation, reduce biodiversity, and thus lead to long-term changes in ecosystem function and structure. For example, forest fragmentation can disrupt energy flow and diversity as forests change from the interior to the edge [10]. Similar ecology theory and findings exist for grassland habitats, where the research emphasizes the need to reduce isolated patches and buffer fragmentation [11]. In addition to their impact on community composition, it has been suggested that fragmented landscapes are more vulnerable to water pollution [12]. Along with recognizing the potential importance of fragmentation, it is also important to quantify the drivers of fragmentation (e.g., urbanization, cropland conversion) as researchers and land managers work towards the ecological sustainability of landscapes in the future.

Globally, temperate grassland ecosystems are the most threatened and least protected of terrestrial biomes [13,14]. In North America, > 50% of temperate grasslands and savannas have been lost, and more than half of the remaining 27 ecoregions in grasslands have experienced high fragmentation [15]. Quantifying these changes is an important prerequisite for understanding the associated ecological processes and implementing grassland conservation. However, compared to forests, fewer studies have concentrated on grassland fragmentation, partially because of the traditional lack of recognition of the conservation value of grasslands [16]. With the development of remote sensing technologies over the past few decades, a growing body of spatially-explicit annual land use datasets has been accumulated at medium spatial resolutions (e.g., 30 m) [17–20]. These datasets have made it possible to detect more detailed temporal and spatial variations in grassland areas at the local and regional scales. For example, Baldi et al. [16] characterized the degree of grassland fragmentation in temperate South America grasslands and analyzed the associated environmental controls based on Landsat TM scenes. Wang et al. [21] quantified grassland land use change and fragmentation in Northeast China from 1954–2000 and attributed the change to agricultural reclamation and salinized wasteland expansion using a time series of topographic maps and satellite images from Landsat Thematic Mapper. Roch and Jaeger [22] evaluated the degree of grassland fragmentation in the Canadian Prairies using the effective mesh size method together with the Crop Inventory Mapping of the Prairies and the CanVec datasets. Wright and Wimberly [23] revealed the rapid rate of grassland conversion to corn/soybean across the U.S. Western Corn Belt based on the National Agricultural Statistics Service Cropland Data Layer. These studies demonstrate the high capacity of time-series satellite data for detecting and characterizing landscape-level details of land use change and fragmented habitats.

The Bluegrass Region in north-central Kentucky possesses some of the most fertile soils in the world [24,25]. The original bluegrass species ecology was open woodland savanna characterized by the abundance of grass understory and individual giant trees [26]. One of the predominant grass species in this region, Kentucky bluegrass (*Poa perenne* L.) was either imported by Eurasian cultures or brought by

European colonists and spread along with the sellers and increase in stock raising, benefiting from ideal soil and climatic conditions [27]. Over the past three centuries, the Bluegrass Region has witnessed gradual changes in land use patterns, with substantial land clearing and a transition to cultivated lands and urban areas in recent decades driven by agricultural exploitation, population growth, and economic development [28–30]. Since the 1960s, population growth has outpaced the provision of urban services, and many Bluegrass Region farms have been converted into housing developments. The percentage of the urban population in Kentucky has burgeoned from 36.8% to 58.4% since the 1950s, although the definition of urban has changed slightly [31]. From 1997–2017, the harvested cropland areas in Kentucky have increased by 12.7%, most of which occurred in western and central Kentucky (https://www.nass.usda.gov/AgCensus/index.php). These changes have threatened the protection of unique natural, scenic, and historic resources in this region. In 2006, the Inner Bluegrass Region was listed as one of the World's Most Endangered Sites by the World Monuments Fund. In addition, Kentucky's Bluegrass Region is characterized by shallow soils and karst geomorphology, which are subject to groundwater contamination. Intensive agricultural activities (such as nitrogen and phosphorus fertilizer use) and suburban sprawl can result in detrimental consequences for regional water security and environmental sustainability. Previous efforts have primarily concentrated on forest loss and forest fragmentation in this region [32], whereas studies that quantify grassland loss and fragmentation across the Bluegrass Region are surprisingly lacking.

In this study, we used the satellite-derived USDA Cropland Data Layer (CDL) to examine the recent shrinkage and fragmentation of grassland in the Bluegrass Region, Kentucky. The specific objectives are (i) to quantify the temporal and spatial patterns in recent land use change across the Kentucky Bluegrass Region; (ii) to characterize the grassland fragmentation using landscape metrics; and (iii) to identify the major drivers responsible for the grassland fragmentation in the Bluegrass Region.

## 2. Datasets and Methods

### 2.1. Description of the Study Area

Our study area is the Bluegrass Region (Figure 1, Figure S1) located in north-central Kentucky, which is renowned worldwide for its abundant Kentucky bluegrass and tall fescue (*Schedonorus arundinaceus* (Schreb.) Dumort.), gently rolling pastures, rich deposits of limestone rock, and horse breeding. Its northern and western parts are bordered by the Ohio River, which provides ample irrigation water for agricultural land. The southern and eastern parts are encircled by a ring of hills known as the Knobs. The Bluegrass Region comprises of the Inner and Outer Bluegrass and extends approximately from 37°28′N to 39°10′N latitude and from 83°25′W to 85°55′W longitude, accounting for approximately one-five of the total area of Kentucky. In this study, the boundary of Bluegrass Region (Figure 1) was derived from the physiographic regions of Kentucky (https://kygeoportal.ky.gov/geoportal), which divides Kentucky into seven sub-regions, i.e., East Coal Field, Bluegrass, Knobs, Western Pennyroyal, Western Coal Field, Cumberland Escarpment, Purchase, and Eastern Pennyroyal (Figure S1). The Bluegrass Region has a humid continental climate, characterized by moderately large variations in temperature and sufficient rainfall [33]. The concentrated rainfall during summer months and warm summer temperatures make this region excellent for crop and pasture plant growth.

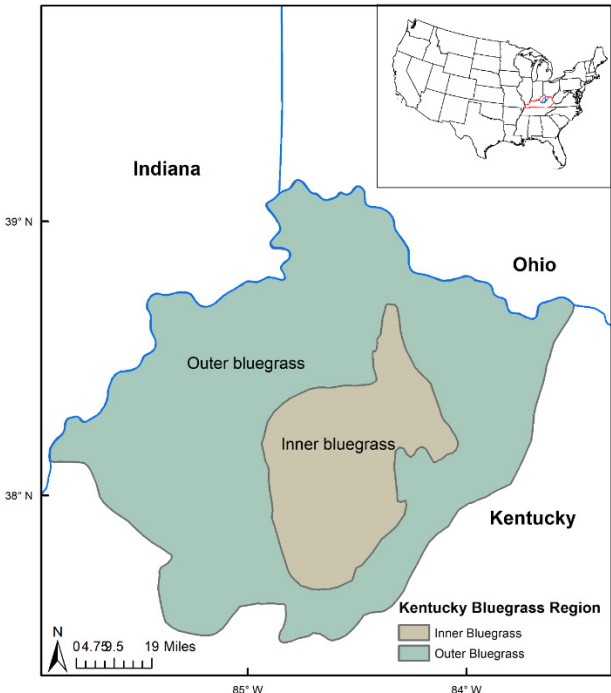

**Figure 1.** Map of the Kentucky Bluegrass Region.

### 2.2. USDA Crop Data Layer

The USDA Cropland Data Layer (CDL) is a high-resolution geo-referenced product that provides multi-year land use and land cover maps at 30 m spatial resolution across the contiguous U.S., with an emphasis on crop-specific distribution [20,34]. The CDL utilizes a number of satellite imageries, including Advanced Wide Field Sensor (AWiFS), Landsat TM/ETM, the Indian Remote Sensing RESOURCESAT-1 (IRS-P6), MODIS, etc. [35]. The CDL program integrates ground survey data with satellite imagery and is more of an "Adjusted Census by Satellite" [36]. The earliest available CDL data were for North Dakota in 1997, and nationwide data has been available since 2008. For Kentucky, the yearly CDL maps are available for the period of 2008–2018, and the overall classification accuracy ranges from 70.3% to 87.1% (https://www.nass.usda.gov/Research_and_Science/Cropland/metadata/meta.php). The CDL data has been widely used to detect field-scale changes over time in land use and land cover for environmental assessments and policy making [23,37–41]. The term "grasslands" describes an ecological region that is highly diverse but difficult to define [42]. It spans a wide variety of grassland types from native/natural grasslands to managed grasslands for forage production to feed livestock. In our fragmentation analysis, we combined all the grass-dominated classes in the Kentucky CDL, including grassland/pasture, other hay/non-alfalfa, sod/grass seed, and fallow/idle cropland, to create a broadly defined grass-dominated class [43].

### 2.3. Grassland Fragmentation Analysis

To examine the spatial patterns in grassland shrinkage and its change over time, we applied a fragmentation model [44,45] based on a moving window algorithm to quantify spatial patterns of grassland fragmentation. This approach has been widely used to examine forest and other natural vegetation and habitat fragmentation analyses using satellite images [46–49]. We firstly generated a binary map of (grassland/no-grassland) and defined two indicators for calculating the fragmentation, grassland area density ($P_f$) and overall grassland connectivity ($P_{ff}$). $P_f$ and $P_{ff}$ were calculated based on a moving window of $5 \times 5$ pixels (approximately 2.25 ha) overlaid over the CDL images. $P_f$ represents the proportion of grassland pixels in the moving window and is calculated by dividing grassland pixels by the total number of land pixels in the window. $P_{ff}$ is overall grassland connectivity,

calculated by dividing the number of adjacent pixel pairs where both pixels are grassland in cardinal directions (*A*) by the number of adjacent pairs that either one or both pixels are grassland in cardinal directions (*B*). For example, in Figure 2e, *A* and *B* are 8 and 1, respectively; therefore, $P_{ff} = 0.125 \approx 0.13$ (rounding to two decimal places). The calculated $P_f$ and $P_{ff}$ were assigned to the central pixel of the moving window. Then the grassland fragmentation for the central pixel was classified into six categories depending on the number of grassland pixels and connectivity between grassland pixels: (interior, $P_f > 0.9$; patch, $P_f < 0.4$; transitional, $0.4 < P_f < 0.6$; edge, $0.9 > P_f > 0.6$ and $P_f - P_{ff} < 0$; perforated, $0.6 < P_f < 0.9$ and $P_f - P_{ff} > 0$; and exterior, all no-data pixels that are outside of the landscape of interest). The calculations are illustrated in Figure 2.

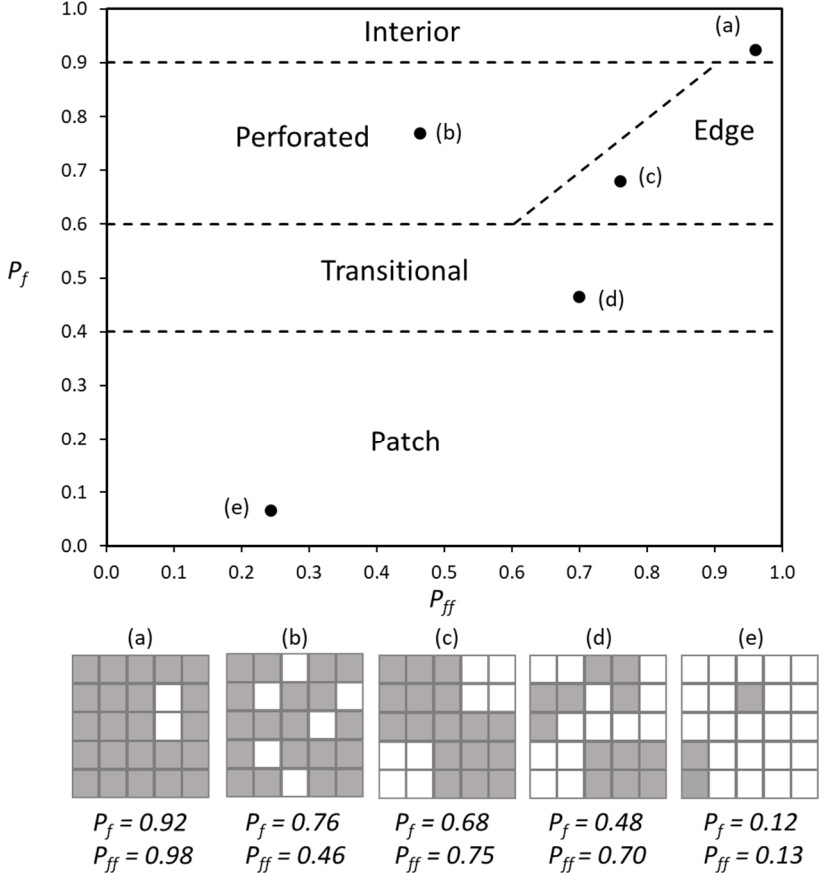

**Figure 2.** Illustration for the calculations of $P_f$ and $P_{ff}$ for various fragmentation categories in a moving window of 5 × 5 pixels. Grassland: grey pixels; no-grassland: white pixels; dashed line: the classification bounds for the various categories of grasslands (adapted from Riitter et al., 2000 [45] and Frate et al., 2015 [49]).

In addition to the moving window analysis, we also identified a set of landscape metrics that capture the complexity of land fragmentation [Table 1]. These landscape metrics have been widely applied to quantify land use/cover change and landscape fragmentation [16,21,48,50]. A total of six landscape metrics were calculated to characterize the grassland fragmentation: (1) the number of patches (PN), (2) patch density (PD), (3) percentage of landscape (PLAND), (4) edge density (ED), (5) patch area mean (Area_MN), and (6) the effective mesh size (MESH). The number of patches (PN) measures the degree of fragmentation of a vegetation type. The metrics of PD is a measure to show the health of a vegetation habitat or an ecosystem. If the PN increases while the areas of the targeted vegetation type decrease, it means fragmentation or dissection occurs [51]. The effective mesh size is measured based on the probability that two pixels chosen randomly in a landscape will be connected, i.e., in the same non-fragmented area of land. The more barriers in the landscape, the lower the

probability that the two points could be connected, i.e., the lower the effective mesh size, the higher the fragmentation level.

**Table 1.** Explanation of landscape metrics used in this study.

| Landscape Index | Abbreviation | Definition | Explanation |
|---|---|---|---|
| Patch density | PD | $Ng/A$ | $Ng$ : number of grassland patches; $A$ : total landscape area (m²) |
| Edge density | ED | $\frac{\sum_{i=1}^{n} Pi}{A}$ | $Pi$ : total length (m) of edge in grassland |
| Percentage of landscape | PLAND | $100\frac{\sum_{i=1}^{n} Ai}{At}$ | $Ai$ : area (m²) of grassland patch $i$ ; $At$ : total landscape area (m²) |
| Number of patches | NP | $n$ | Number of grassland patches |
| Patch area mean | AREA_MN | $AREA_{MN} = \sum_{i=1}^{n} A_i/n$ | Mean Grassland Patch Area |
| Effective mesh size | EFMS | $EFMS = \frac{\sum_{i=1}^{n} (Ai)^2}{At}$ | |

The spatial pattern analysis software FRAGSTATS 4.2 [52] and a Python toolbox for zonal landscape structure analysis [53] on the ArcGIS platform were used in this study to calculate the landscape metrics and conduct fragmentation analysis. The extension of the patch analyst facilitates the spatial analysis of landscape patches and the modeling of attributes associated with patches [21]. The Zonal Metrics toolbox allows the calculation of landscape metrics for user-defined zones.

## 3. Results

### 3.1. Recent Changes in Grassland Across the Bluegrass Region

The current land use distribution in the Bluegrass Region is shown in Figure 3a. Grasslands are concentrated in the Inner Bluegrass Region, the south and east of the Outer Bluegrass Region, with some distributed in the west of the study area. Forests are mainly distributed in the north and west of the study area, while the built-up areas are mainly concentrated in the center of the Inner Bluegrass Region, the northernmost and the far west of the study area, the metro areas within Kentucky of Lexington, Cincinnati, and Louisville. Our results suggest that grasslands significantly decreased across the Kentucky Bluegrass Region (Figure 3b) from 2008–2018. In total, the grassland area has decreased by 14.4% since 2008. Land conversions from grassland have been widespread, with major shrinkage occurring in the west and northeast of the Outer Bluegrass Region and relatively minor grassland conversion in the Inner Bluegrass Region (80% vs. 20%) (Figure 3b). Land conversions to grassland were scattered in the southeast and northwest of the Outer Bluegrass Region and south of the Inner Bluegrass Region. Further analysis shows that recent land use change across the Bluegrass Region was characterized by grassland decline, cropland expansion, forest increases, and suburban sprawl. Croplands, forests, urban, shrublands contributed to the net decrease in grasslands by 46%, 43.9%, 4.5%, and 4.3%, respectively (Figure 3c).

The land use transition matrix showed that, during the study period, approximately 86,210 ha, 79,510 ha, 9,890 ha, and 7,110 ha of grassland were converted into croplands, forests, developed areas, and shrublands, respectively (Table 2). Only a small part of the decrease in grassland was offset by land conversion from other land use types into grasslands. Croplands, forests, and developed areas accounted for 58%, 30%, and 10% of the increase in grasslands, respectively, with others from shrublands, water bodies, and barren land. It should be noted that lawns converted from developed areas were identified as grassland in the CDL data; typically, this is managed grass space, and it makes a small contribution to the mitigation of grassland fragmentation in the study area.

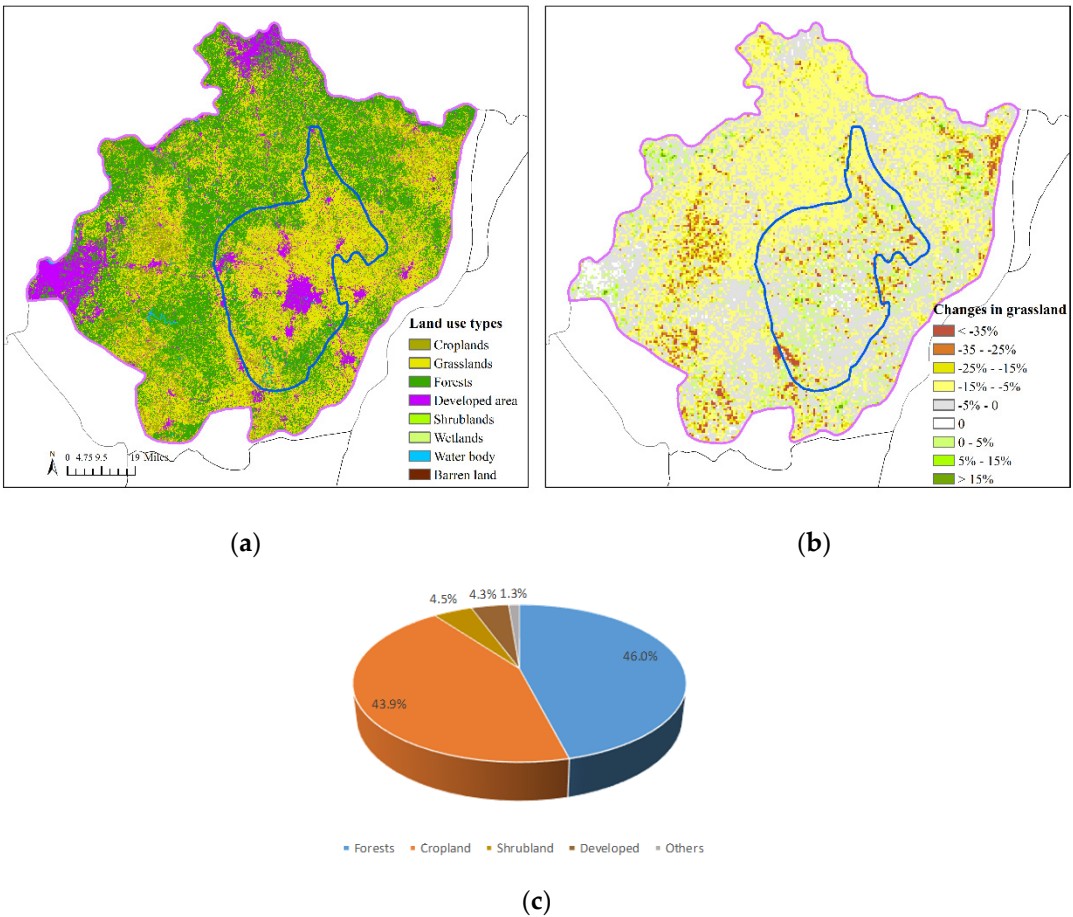

**Figure 3.** Spatial patterns in land use (**a**) and grassland conversions (**b**) and contributions of different land use types to the grassland shrinkage across the Bluegrass Region (**c**) (the CDL map was reclassified into major land use types).

**Table 2.** Land conversion matrix for the Bluegrass Region from 2008 to 2018 (unit: 1000 ha).

| 2008 | 2018 | | | | | | | |
|---|---|---|---|---|---|---|---|---|
| | Cropland | Grassland | Forests | Developed | Shrubland | Wetland | Water | Barren |
| Cropland | 41.82 | 20.1 | 1.52 | 0.48 | 0.05 | 0 | 0.05 | 0.08 |
| Grassland | 86.21 | 859.24 | 79.51 | 9.89 | 7.11 | 0.03 | 1.48 | 0.99 |
| Forests | 2.02 | 10.21 | 814.17 | 2.07 | 1.38 | 0.57 | 1.1 | 0.24 |
| Developed | 1.58 | 3.36 | 0.94 | 236.81 | 0.01 | 0.02 | 0.11 | 0.51 |
| Shrubland | 0.01 | 0.29 | 1.62 | 0 | 4.71 | 0 | 0.02 | 0 |
| Wetland | 0 | 0 | 0.31 | 0 | 0 | 0.18 | 0.02 | 0 |
| Water | 0.02 | 0.47 | 1.05 | 0.06 | 0.1 | 0.05 | 27.13 | 0.26 |
| Barren | 0.01 | 0.11 | 0.03 | 0.12 | 0 | 0 | 0.02 | 0.19 |

### 3.2. Analysis of Grassland Fragmentation

Our results suggest that the total grassland patch area increased slightly during the period of 2008-2018 and accounted for 7.6% when using a 5 × 5 window for fragmentation mapping (Figure 4). The increase in the total patch area means that more grassland habitats were transformed into a number of isolated patches. Areas for all other categories shrank during the study period. The interior grassland category, which represents the most expansive ecosystem in the study area, saw a 5% (approximately 0.5 M ha) decrease over the past 11 years. Perforated grassland decreased by 9% (approximately 0.8 M ha) (Figure 4). This process is usually regarded as the first stage of vegetation fragmentation and involves land conversion from grassland to other land use types. The increased patch area and decreased interior and perforated grasslands show that the Bluegrass Region has seen a

sharp shrinkage in intact grassland ecosystems and has become more fragmented due to the land use changes during the period of 2008–2018.

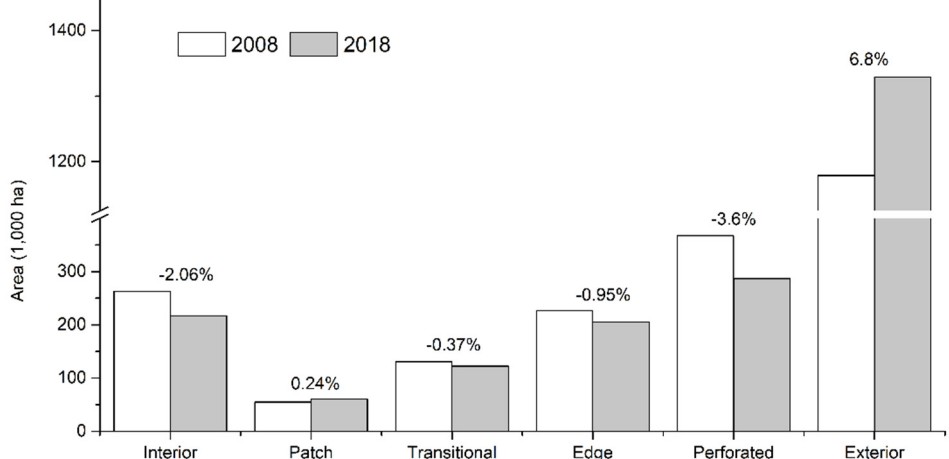

**Figure 4.** Changes in areas of various fragmentation categories during 2008–2018.

The edge and transitional grassland areas experienced slight decreases during the study period (Figure 4). However, our further analysis suggested that most of the reductions in transitional areas were from conversions to the exterior categories, i.e., the highest level of grassland fragmentation (32.7% of the total transitional areas), followed by conversion to the patch areas (17.4%). Similarly, most of the decreases in edge areas were from the conversions to the exterior (14.1%) of the total edge areas and transitional (12.5%) categories (Figure 5c,d). Our results also suggested that approximately 16.10% and 10.20% of interior grasslands were degraded to the perforated and patch areas, respectively (Figure 5a). Approximately 14.7% and 13.5% of the perforated areas had converted to the exterior and transitional areas, respectively, identified as more fragmented categories, although about 8.4% changed to interior grasslands (Figure 5b). For the patch areas, approximately 55.8% and 3.9% had converted to the exterior and transitional categories (Figure 5e). When investigating spatial patterns in grassland fragmentation, our results showed sharp reductions in the interior grasslands in the northern and southwestern Inner Bluegrass Region, and the western and northeastern Outer Bluegrass Region (Figure 6a, Table S1). Further analysis showed that relatively more interior grasslands were converted to the categories of perforated and exterior in parts of the western and eastern Outer Bluegrass Region and northern Inner Bluegrass Region (Figure 6b,f).

Grassland fragmentation patterns in the Bluegrass Region were also revealed by using several landscape metrics (Table 1, Table 3). Our results suggested that the percentage of landscape (PLAND) decreased from 61.55% to 55.4% and from 43.49% to 36.5% in the Inner and Outer Bluegrass Regions during 2008–2018, respectively. The number of grassland patches (NP) increased from 13,654 to 17,543 and from 95,965 to 109,853 in the Inner and Outer Bluegrass Regions, respectively. For the entire Bluegrass Region, the PLAND decreased from 47.01% to 40.24% during the period of 2008–2018. The edge density (ED) decreased slightly, which was consistent with our analysis based on the vegetation fragmentation model. The effective mesh size (EFMS), representing the degree of landscape fragmentation in a system, indicated obvious decreases in the Bluegrass Region, with relatively lower decreases (e.g., higher fragmentation degree) in the Outer Bluegrass Region.

**Table 3.** Landscape metrics of grassland in the Bluegrass Region during 2008–2018.

| Period | | PLAND | NP | PD | ED | AREA_MN | MESH |
|---|---|---|---|---|---|---|---|
| 2008 | Inner BR | 61.55 | 13654 | 2.93 | 85.56 | 21.02 | 4663.71 |
| | Outer BR | 43.49 | 95065 | 5.42 | 98.83 | 8.03 | 2675.22 |
| | Entire BR | 47.01 | 108338 | 4.88 | 95.59 | 9.64 | 6629.84 |
| 2018 | Inner BR | 55.4 | 17543 | 3.76 | 85.86 | 14.72 | 1655.28 |
| | Outer BR | 36.5 | 109853 | 6.26 | 89.94 | 5.83 | 936.54 |
| | Entire BR | 40.24 | 126874 | 5.71 | 88.66 | 7.04 | 1816.58 |

Note: Inner BR: Inner Bluegrass Region; Outer BR: Outer Bluegrass Region; Entire BR: Entire Bluegrass Region.

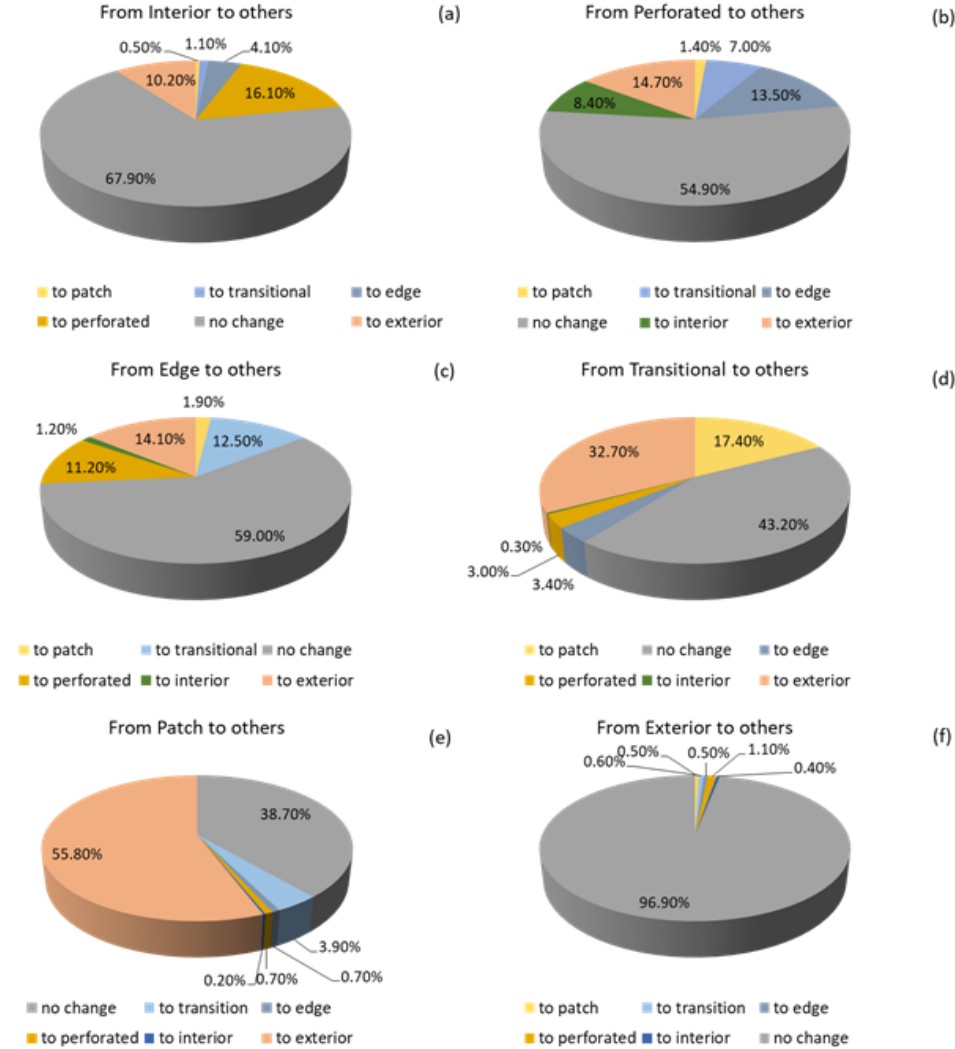

**Figure 5.** Changes between fragmentation categories during 2008–2018: (**a**) from interior to others, (**b**) from perforated to others, (**c**) from edge to others, (**d**) from transitional to others, (**e**) from patch to others, and (**f**) from exterior to others.

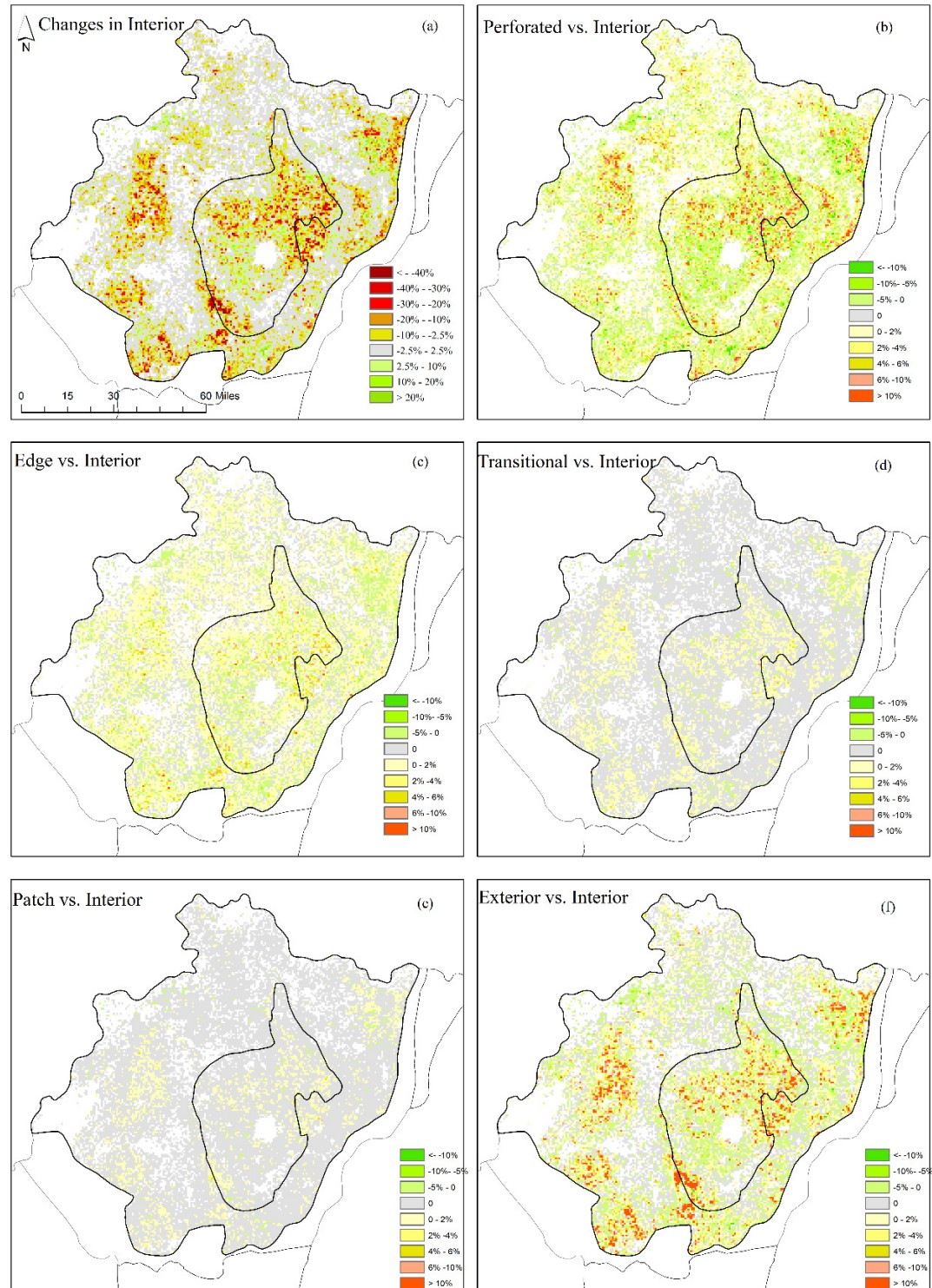

**Figure 6.** Spatial patterns in conversions between interior grassland and other fragmentation categories during 2008–2018: (**a**) changes in interior, (**b**) perforated vs. interior, (**c**) edge vs. interior, (**d**) transitional vs. interior, (**e**) patch vs. interior, and (**f**) exterior vs. interior.

## 4. Discussion

### 4.1. Grassland Shrinkage and Fragmentation and Driving Factors

This study identified a sharp shrinkage in grassland and fragmentation in the Bluegrass Region over the recent decade. Although recent grassland losses have been widely reported in the U.S., these

studies have mainly focused on the Midwest U.S. and the Great Plain regions [23,43,54,55]. For example, Wright and Wimberly [23] estimated a net decline of 530,000 ha in grass-dominated land cover across the Western Corn Belt during the period of 2006–2011. Based on our estimate using the CDL data, we found that this grassland loss was about 5.7% of total grassland areas in this region in 2006. Most of the grasslands were converted to corn and soybeans at a rate of 5%–30% across 560 m pixel-levels. Reitsma et al. [35] reported that 730,000 ha of grasslands (equivalent to 6.87% of the grassland area) were changed to cropland and 250,000 ha of croplands (4.2%) were converted to grasslands in South Dakota during 2006–2012. In the Dakota Prairie Pothole Region, the grassland area was 9 % lower in 2012 than in 2006, about 20% of which was converted to corn and soybeans [54]. Our estimated grassland losses (14.4%) over the study period in the Bluegrass Region were even greater than the above mentioned regions, which have been well known for their recent grassland reduction, despite a longer period covered by this study. The contribution of cropland to the loss of grassland was around 46%, which is higher than that in the Dakota Prairie Pothole Region [54]. Furthermore, grassland losses were mainly attributed to cropland expansion in these regions, in contrast with the Bluegrass Region, where afforestation and urbanization (which occurred mainly in peri-urban areas in spite of a lower overall contribution) were also major influencing factors. In the U.S., grasslands have been the most common source of new croplands [56], which has significant implications for wildlife habitat, biodiversity, and biogeochemical cycles at broad-scales [22,54]. However, in the U.S., most researchers have focused on major agriculture regions such as the Midwest, whereas grassland use change in other areas (e.g., Kentucky Bluegrass Region in this study) have largely been overlooked. Moreover, grassland shrinkage and fragmentation may have greater effects on environmental sustainability in the Bluegrass Region, given its widespread karst geomorphology, especially in the Inner Bluegrass Region. In the long term, synergies and tradeoffs between regional food security, economic flourishment, and natural habitat protection should be considered. More studies are needed to quantify the multiple effects of grassland use change in the Kentucky Bluegrass Region.

As for grassland fragmentation, our estimated large decreases in the interior grassland category suggest that the intact grasslands across the Bluegrass Region are at high risk of further fragmentation. As estimated by the World Wildlife Fund [57], overall, roughly 54% of the Great Plains is still intact grassland. This number is higher than the Bluegrass Region (<30%) if we define the interior grassland category as intact grassland. Most of the decreases in interior grasslands were due to their conversion to perforated and edge grasslands or other land use types. Perforating processes create holes in otherwise contiguous grassland habitat, with the significant potential to develop into higher fragmentation categories (such as transitional and edge, Figure 5b). Similar situations happened in other fragmentation categories (Figure 5c–e). As grassland fragmentation varies over the region, it is expected there will be parallel effects on ecosystem functioning and structure, with significant implications for water, energy, and biogeochemical processes [16].

*4.2. Potential Effects on the Biogeochemical and Hydrological Cycles*

Grasslands store most of their carbon below ground (up to 98%) and may accumulate substantial soil carbon over time [58]. Some studies suggested that U.S. grasslands served as a net carbon sink during the first decade of the 20th century, although there are large discrepancies in various studies [59–61]. When native or long-term grasslands are converted to other land use types, carbon is expected to be released into the atmosphere [61–63]. Specifically, the conversion from grasslands to croplands may result in abrupt soil carbon loss and large $N_2O$ emission to the atmosphere, which represents the biggest threat to greenhouse gas mitigation [64]. On the other hand, afforestation from grasslands is widely regarded as an important means to obtain carbon benefits [65], although associated carbon benefits are highly dependent on local climate and soil conditions. Our results show that crop expansion, afforestation, and urban sprawl contributed to most of the loss in grassland shrinkage. Quantifying the associated carbon consequences needs to incorporate inventory and a spatially-explicit ecosystem/landscape modeling approach.

Grasslands provide a significant hydrological buffer, and landscape structure represents one of the most important factors influencing runoff and nutrient leaching in watersheds [66–68]. Some studies have suggested that fragmented landscapes tend to be more vulnerable to water pollution [12], and areas that are more urbanized and fragmented may have increased nutrient and heavy metal contamination in their groundwater [69–71]. Intact forests and grasslands have the ability to sequestrate nutrients in surface runoff and reduce the concentrations of nitrogen and phosphorus pollutants in water bodies [72]. Shi et al. [73] studied the relationship between grassland/forest areas and water quality data over five riparian sites and found that grasslands and forests explain 14.5% and 25.5% of the variation in water quality in the dry and wet seasons in Shaanxi Province, China. In the Kentucky Bluegrass Region, a site-level study demonstrated that nitrate and ammonium losses declined significantly once turfgrass cover was established [74]. In the northeastern Bluegrass Region, nitrate and orthophosphate concentrations were found to be significantly higher in agriculture-dominated watersheds, and the total suspended solids, turbidity, temperature, and pH, were relatively higher in the urban and mixed watersheds [75]. All these studies show that built-up areas, grasslands, and forests can exert significant influences on a number of water quality indicators. As an area where shallow soils and karst geology are widespread, the Kentucky Bluegrass Region might be more sensitive to natural habitat losses and landscape fragmentation. When comparing the land use map for the Bluegrass Region (Figure 2, Figure 5) with the Kentucky groundwater sensitivity region [76], we found that a large portion of grassland decreases occurred in areas that were rated as moderately groundwater sensitive in the Outer Bluegrass Region (Figure 3, Figure S2). Even more concerning, the loss of interior grassland in the northern Inner Bluegrass Region was in areas rated as highly to extremely sensitive and were highly karstic. This situation might present a challenge for regional water resources management. Further efforts are required to quantify the associated effects and inform policymakers to mitigate environmental side effects.

### 4.3. Landscape Fragmentation and Local Culture

The local culture in the Bluegrass Region has been influenced not only by people's propensities but also by natural conditions such as vegetation, climate, etc. [26]. Central Kentucky is known as "horse country" due to its unique and favorable environment for grass growth. The bluegrass species and related genera on limestone-derived soils across this region are rich in calcium and phosphorus, which builds strong bones in horses [27]. The Kentucky Equine Survey in 2012 reported that Kentucky's equine industry had a total economic impact of approximately $3 billion and created 40,665 jobs. The recent grassland shrinkage and fragmentation have jeopardized the sustainability of the horse industry and put the co-existence of human culture and the natural environment at risk [77]. In the Inner Bluegrass Region, several land use policies have been implemented to maintain larger grassland tracts and minimize the side effects of suburban sprawl. For example, Woodford county enforces a 40-acre minimum farm size (acreage minimums have been established in most Inner Bluegrass counties), and Lexington/Fayette County limits development outside their established economic development zone. Wise land use planning that carefully considers the impacts of land use on socioeconomic sectors is needed.

### 4.4. Uncertainties and Future Needs

Quantifying recent land use change and grassland fragmentation is essential for formulating appropriate conservative management strategies to promote the health and sustainability of the Bluegrass Region ecosystem. The quality of the estimates derived from this study depends on the land use maps and approaches being used. In this study, our land use change and grassland fragmentation analyses were based on the USDA Cropland Data Layer, which predicted land use change using a number of satellite products at high resolutions. Uncertainties that accumulate and propagate in the classification process ultimately affect the accuracy of the classification, and thus, the reliability of the land use and grassland fragmentation analyses. The grassland categories are typically regarded as

the weakest feature of the CDL products [78]. The accuracy of grassland classification had a wide range and is highly dependent on the dominant vegetation in a given region, e.g., the grassland accuracies were lower in the cropland-dominated areas, and cropland accuracies were lower in the grassland-dominated areas [35]. For example, Reitsma et al. [79] evaluated the 2006-2012 CDL maps for South Dakota against ground-collected data and found that the accuracies for grassland ranged from 75.2% to 95.2%, except for southeastern areas where grasslands were sparsely distributed. Considering that both South Dakota and the Kentucky Bluegrass Region are grassland-dominated areas, it is reasonable to deduce that our estimated decreases in grassland (14.4%) fall within a similar accuracy range. In future studies, multi-data fusion for detecting and quantifying should be used for providing more detailed insights into regional land use change with higher mapping accuracies [78]. Increasingly available optical and microwave data (≤30m) and some commercial satellite data (≤5m), in conjunction with high-performance computing techniques, are expected to enhance monitoring of land use change and to separate sub-categories of grasslands, for example, pasture versus hay. On the other hand, most landscape fragmentation studies have focused on forests [45,48,80] and were recently extended to other vegetation habitats [81–83], with relatively fewer studies on grasslands [16]. The indicators or metrics from these studies used for estimating vegetation fragmentation are highly varied. It has been suggested that the analysis of interior, perforated, transitional, and edge vegetation can be very helpful for identifying current fragmentation and predicting potential change trends [48]. Roch and Jaeger [22] made a leading effort to use the effective mesh size (EFMS in this study) and found it is highly suitable for quantifying grassland fragmentation. In this study, both the fragmentation model and landscape metric approaches were used to detect the grassland fragmentation in the Bluegrass Region. However, further efforts are needed to investigate the behavior of various landscape metrics and quantify the suitability of different approaches to detect fragmentation in various vegetation habitats. In addition, in some cases, grasslands are part of farming practices for soil fertility recovery from intensive crop management [54]. Because we were limited by the spatial resolution of the CDL data, we did not examine the fragmentation due to roads and the effects of suburban yards and highway, transmission line, and pipeline rights-of-way, which requires a combination of high-resolution satellite images. Quantifying grassland use change and fragmentation requires a full evaluation of all land uses in which grasslands are involved.

## 5. Conclusions

This study shows that the grassland ecosystem in the Bluegrass Region, Kentucky, has experienced significant shrinkage and has become more fragmented over the past decade. The grassland area has decreased by 14.4% over the period of 2008–2018, with a decline of 5% in interior grassland, which has been driven by cropland expansion, afforestation, and suburban sprawl. The effective mesh sizes, a metric for comparing the degree of fragmentation of landscapes, have obviously decreased in both the Inner and Outer Bluegrass Regions. Our study highlights the environmental pressures from urban growth and development and cropland expansion, which jeopardizes the provision of crucial ecosystem services, including but not limited to water purification, biodiversity, and wildlife habitat. Therefore, we call for more intensive monitoring and further conservation efforts to preserve the unique ecosystem in the Bluegrass Region, which has both local and regional implications for climate mitigation, carbon sequestration, diversity conservation, and culture protection.

**Supplementary Materials:** The following are available online at http://www.mdpi.com/2072-4292/12/11/1815/s1, Figure S1: Groundwater sensitivity regions of the Bluegrass Region (modified from the map of Groundwater Sensitivity Regions of Kentucky), Figure S2: Map of physiographic regions in Kentucky, Table S1: Conversions between various fragmentation categories between 2008–2018.

**Author Contributions:** Data curation, B.T. and Y.Y.; Formal analysis, B.T. and Y.Y.; Funding acquisition, W.R.; Investigation, W.R.; Methodology, J.Y.; Project administration, W.R.; Writing—original draft, B.T.; Writing—review and editing, J.Y., R.S., J.F., A.C.R., J.L. and W.R. All authors have read and agreed to the published version of the manuscript.

**Funding:** This study was supported by the NASA Kentucky Space Grant Consortium and EPSCoR Programs (NNX15AR69H).

**Acknowledgments:** This study was supported by the NASA Kentucky Space Grant Consortium and EPSCoR Programs (NNX15AR69H), a collaboration project between UKY and NASA GISS. Alex C. Ruane acknowledges support from the NASA Earth Science Division through the NASA Goddard Institute for Space Studies Climate Impacts Group. We thank Rebecca L. McCulley for the comments and suggestions on the manuscript.

**Conflicts of Interest:** The authors declare no conflict of interest.

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
