# Peer review of "Recent Shrinkage and Fragmentation of Bluegrass Landscape in Kentucky"

_remotesensing, doi:10.3390/rs12111815_

Round 1
Reviewer 1 Report
This paper takes problem of long-term changes of land cover, in particular grassland in Kentucky. The proposed methodology with GIS tools and detailed spatial data are approbated and often used in similar studies. Paper structure is correct and the obtained results are well argued.
Some detailed comments are provided in the text (enclosed pdf).

Reviewer 2 Report
The manuscript presents a good analysis of land cover changes in the Bluegrass Region. The main conclusion, viz. that the grassland shrinks, is supported by a careful investigation. The only major shortcoming of the article that I can see is the somewhat vague discussion of the uncertainties. I appreciate the discussion in section 4.4, but it lacks quantitative statements. The main result is that the grassland area has decreased by 14.4% - is the uncertainty 0.1%, 1% or 10%? At least an estimate of the order of magnitude with a short explanation why this value was chosen should be added, with other words: Replace the word "slightly" in line 393 with a number.
Minor corrections:
Lines 85: explain the abbreviation "TM". Readers not familiar with Landsat might not know it.
Line 49/142: The abbreviation "USDA" appears already in line 49, but is explained only in line 142.
Line 150: accuracy of what?
Line 168: ha is a unit of area, delete the 2.
Line 193: probability of what?
Table 1: For consistency, the two densities should have the same units. There are two Table 1s. Explain the abbreviation "NCP". There is no text to footnote "1".
Figure 3 (a) and (b): Letters and numbers are difficult to read.
Lines 228, 273: There are no Tables 2 or 3.
Line 294: Why is 14.4% (Bluegrass Region 2008-2018) greater than 30% (Corn Belt 2006-2011) or 40% (Pothole Region 2011-2012)?
Line 306: This is not true - regional, economic flourishment can be at the cost of habitat losses, because there are other economies than agriculture, and the land losses could be compensated elsewhere. This sentence is rather an opinion than a scientific fact.
Line 380: What are "Needsands"?
Author Response
Please see thee attachment.

Reviewer 3 Report
This is an interesting study, presented in a well written manuscript employing well established landscape analysis methods. It is true that landscape dynamics studies focusing on grasslands are needed, given the significant importance of those ecosystems in maintaining biodiversity and several ecosystem services, and their dependence on management for their sustainability. The manuscript reads well, the use of English is academic and I have only spotted a couple of minor mistakes which I have highlighted in the text
The introduction presents all the background information which allow the reader to follow the manuscript. It concludes with a clear set of objectives which are addressed in the manuscript. The methods are described in details, apart from a couple of points that I have highlighted in the text, which allow their transferability to other areas where similar data exist. The authors employ two approaches to investigate the grassland dynamics in the area. The first rely on a moving window and two metrics, while the second on a landscape analysis using landscape indices. Both approaches are scientifically sound, well established and adequately described in the literature, while they both result on the same observations. Grasslands are becoming less abundant and more fragmented as a result of a polarised pressure, primarily from agriculture and forest encroachment.
The results are presented with a good combination of text, tables and figures and apart from the mistake with table 2, which I have highlighted in the text, and a few more clarifications that are necessary, they are easy to follow.
In the discussion section there is nothing to disagree on and it is nicely written. However, I believe the authors spend more than the necessary space to discuss the implications of grassland loss, which are not studied here, the limitations of the study and the future needs. I believe the authors should tip the balance slightly, in favour of the discussion focusing on the results, which currently occupies only one third of the discussion.
The main limitation of the study is the one already acknowledged by the authors and I appreciate that. The accuracy of the primary data used brings to the whole study a great degree of uncertainty. 70% accuracy is probably way to low for such a study. However, the same data set or datasets with similarly low accuracies have been widely used in the literature in similar studies. Furthermore, the results obtained are the anticipated, so despite this limitation and given that it is already acknowledged by the authors I believe the study deserves to be published.
Having said all these, one would thing that the “reject” button was clicked on by mistake. If I was reviewing this manuscript for the journal “Sustainability” or “Land”, my recommendation would clearly be “accept after minor revision”. However, for the journal “Remote Sensing” it has to be a rejection simply because this is not a remote sensing manuscript. There is not a single remote sensing method employed in the study, nor a single remote sensing dataset. The fact that the mapping products used were derived using EO data, as most mapping products do, is not enough to classify this study as a study in the field of remote sensing. I will sincerely recommend the authors to resubmit their manuscript in a more appropriate journal, like the ones I am mentioning above, and I would be most happy to review the manuscript again and recommend minor revision. I am sorry for this recommendation, and I would like to make clear once again that I do believe that the study should be published but not in Remote Sensing.

Author Response
Please see thee attachment.

Round 2
Reviewer 3 Report
Dear authors
As i wrote in my first review i had no major concerns about the manuscript per se. My main concern was that i do not believe this is a remote sensing manuscript. I still believe the same and i have a very long experience in remote sensing. But in the end of the day it is not my decision to make on whether the manuscript fits in the journal. It is the editor who decides on that and I support his/her decision to proceed with the manuscript. My other minor concerns have been addressed in the revised version of the manuscript so i am happy to suggest acceptance of the manuscript in its present form.